# Changes in sick notes associated with COVID-19 from 2020 to 2022: a cohort study in 24 million primary care patients in OpenSAFELY-TPP

Andrea L Schaffer ,[1] Robin Y Park,[1] John Tazare,[2] Krishnan Bhaskaran,[2] Brian MacKenna,[1] Spiros Denaxas,[3,4,5] Iain Dillingham,[1] Sebastian C J Bacon,[1] Amir Mehrkar,[1] Christopher Bates,[6] Ben Goldacre,[1] Felix Greaves,[7,8] John Macleod,[9] The OpenSAFELY Collaborative, National Core Studies Collaborative, Laurie A Tomlinson,[2] Alex Walker[1]

**Correspondence to**
Dr Andrea L Schaffer;
andrea.schaffer@phc.ox.ac.uk

## ABSTRACT

**Objectives** Long-term sickness absence from employment has negative consequences for the economy and can lead to widened health inequalities. Sick notes (also called 'fit notes') are issued by general practitioners when a person cannot work for health reasons for more than 7 days. We quantified the sick note rate in people with evidence of COVID-19 in 2020, 2021 and 2022, as an indication of the burden for people recovering from COVID-19.

**Design** Cohort study.

**Setting** With National Health Service (NHS) England approval, we used routine clinical data (primary care, hospital and COVID-19 testing records) within the OpenSAFELY-TPP database.

**Participants** People 18–64 years with a recorded positive test or diagnosis of COVID-19 in 2020 (n=365 421), 2021 (n=1 206 555) or 2022 (n=1 321 313); general population matched in age, sex and region in 2019 (n=3 140 326), 2020 (n=3 439 534), 2021 (n=4 571 469) and 2022 (n=4 818 870); people hospitalised with pneumonia in 2019 (n=29 673).

**Primary outcome measure** Receipt of a sick note in primary care.

**Results** Among people with a positive SARS-CoV-2 test or COVID-19 diagnosis, the sick note rate was 4.88 per 100 person-months (95% CI 4.83 to 4.93) in 2020, 2.66 (95% CI 2.64 to 2.67) in 2021 and 1.73 (95% CI 1.72 to 1.73) in 2022. Compared with the age, sex and region-matched general population, the adjusted HR for receipt of a sick note over the entire follow-up period (up to 10 months) was 4.07 (95% CI 4.02 to 4.12) in 2020 decreasing to 1.57 (95% CI 1.56 to 1.58) in 2022. The HR was highest in the first 30 days postdiagnosis in all years. Among people hospitalised with COVID-19, after adjustment, the sick note rate was lower than in people hospitalised with pneumonia.

**Conclusions** Given the under-recording of postacute COVID-19-related symptoms, these findings contribute a valuable perspective on the long-term effects of COVID-19. Despite likely underestimation of the sick note rate, sick notes were issued more frequently to people

## STRENGTHS AND LIMITATIONS OF THIS STUDY

⇒ Our data come from a large, representative sample of more than 40% of primary care patients in England.

⇒ To better understand sick note patterns, we identified COVID-19 cohorts from three different pandemic time periods, with both contemporary and historical comparators, to account for changes in preventive measures and disruptions to working patterns that would have impacted on receipt of sick notes over time.

⇒ We relied on three different methods for identifying COVID-19 cases (positive PCR or lateral flow test test, primary care diagnosis, hospitalisation), but there is still bias in who gets tested or seeks care for COVID-19 which will have changed as the pandemic evolved. It is unclear how this would correlate with the likelihood of requesting or requiring a sick note.

⇒ We could not identify whether people were participating in the workforce and included everyone of working age in the denominator. It is, therefore, likely that the true absolute sick note rate is higher than estimated.

with COVID-19 compared with those without, even in an era when most people are vaccinated. Most sick notes occurred in the first 30 days postdiagnosis, but the increased risk several months postdiagnosis may provide further evidence of the long-term impact.

## BACKGROUND

In primary care, a doctor may issue a sick note (commonly referred to as a 'fit note' in the UK) after the first 7 days of sickness absence if the doctor assesses that the patient's health affects their fitness for work. In 2021–2022, over 11 million sick notes were issued in England.[1] Long-term sickness absence from employment has negative consequences for

the broader economy and can lead to widened health inequalities, financial insecurity and reduced social participation.[2] Improving the health and productivity of the population and reducing welfare benefit claims are important policy objectives.[3 4]

The Office of National Statistics (ONS) estimated that 82% of the English population had been infected with SARS-CoV-2 by November 2022.[5] Additionally, an estimated 1.9 million people self-reported as experiencing long COVID symptoms in the UK in the 4 weeks prior to March 2023, with two-thirds experiencing symptoms for at least 1 year.[6] The prevalence of long-term symptoms was greatest in people aged 35–69 years, women and people living in more deprived areas.[6] Previous UK research has also shown that receipt of sick notes varies by demographics, with women and people in manual and service occupations more likely to be issued a sick note.[4] To date, there has been limited research on sick notes issued to patients recovering from COVID-19 in England.

The pandemic has evolved over time, both in the number of cases, disease severity and demographic groups most affected.[7] Given the risk of long-term symptoms following COVID-19 ('long COVID'), it is of public health interest to quantify the impact of infection and recovery on the workforce and how that has changed over the pandemic. However, long COVID is heterogeneous and difficult to define.[8 9] Furthermore, coding of long COVID in primary care is very low[10] and cannot be relied on to identify people suffering from persistent symptoms. Therefore, our objectives were to (1) describe the demographic and clinical characteristics of people given a sick note following a documented SARS-CoV-2 infection or COVID-19 diagnosis; (2) determine how the sick note rate varies over time postdiagnosis and (3) quantify the difference in sick note rate in people with SARS-CoV-2 infection or COVID-19 diagnosis compared with the general population and people hospitalised with pneumonia.

## METHODS
### Study design
We conducted an observational cohort study using general practice primary care electronic health record (EHR) data from primary care practices in England.

### Data source and data sharing
We used primary care data from approximately 40% of the English population currently registered with general practitioner (GP) surgeries using TPP SystmOne software. All data were linked, stored and analysed securely using the OpenSAFELY platform, https://www.opensafely.org/, as part of the National Health Service (NHS) England OpenSAFELY COVID-19 service. Data include pseudonymised data such as coded diagnoses, medications and physiological parameters. No free text data are included. All code is shared openly for review and reuse under MIT open licence (https://github.com/

opensafely/long-covid-sick-notes). Similarly, pseudonymised datasets including ONS registered deaths, hospital episode statistics (HES) and second generation surveillance system (SGSS) COVID-19 test results are securely provided to TPP and linked to primary care data. Detailed pseudonymised patient data are potentially reidentifiable and not shared.

### Study population
We included adults 18–64 years registered with one GP for at least 1 year prior to their index date with information on age, sex, index of multiple deprivation (IMD) and the sustainability and transformation partnership region (Sustainability and Transformation Partnership (STP) an NHS administrative region). This age range was selected to represent people most likely to be in the workforce. From this source population, we identified three cohorts with recorded SARS-CoV-2 infection between 1 February and 30 November in each of 2020, 2021 and 2022 (hereafter referred to as the 'COVID-19 cohorts'). We used the period February to November (instead of full years) to allow a wash-out period between years.

The COVID-19 cohorts were identified through three routes: having a recorded positive test for SARS-CoV-2 based on SGSS data, which captures both PCR and lateral flow tests (LFT); having a probable diagnostic code for COVID-19 in primary care records or being hospitalised with a primary or secondary diagnosis for COVID-19 (International Classification of Disease (ICD)-10 codes U07.1 or U07.2) identified from Hospital Episode Statistics (HES). The index date was the earliest of these events. As comparators, we identified contemporary and one historical (2019) general population cohort who were frequency matched in age, sex and STP to the COVID-19 cohorts. The 2019 general population cohort was matched with the 2020 COVID-19 cohort. For the comparator cohorts, the index date was randomly assigned and randomly distributed over the study period.

We also performed a secondary analysis among the subset of people hospitalised with COVID-19. Here, the index date was the date of admission. We compared these individuals with people hospitalised with pneumonia between 1 February 2019 and 30 November 2019. We chose pneumonia as a comparator to provide context, as it is a common, serious respiratory disease with which clinicians and policy-makers will be familiar. The pneumonia cohort was identified using the following ICD-10 codes in any diagnosis position: B01.2, B05.2, B20.6, B25.0, J10–J18, J85.1, U04. No contemporary pneumonia comparator cohort was included due to the potential for misclassification with COVID-19. People could be included in multiple cohorts if they met the inclusion criteria. A figure depicting the 11 cohorts is in figure 1.

### Study measures
In England, employees can self-certify for the first 7 days of sickness; after this, they can receive a Statement of Fitness for Work (also called a fit note or sick note) from their

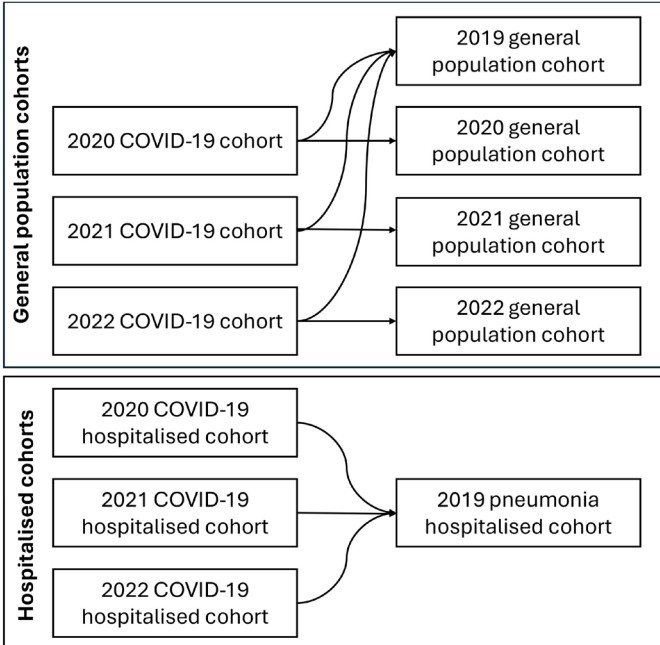

**Figure 1** Depiction of 11 cohorts and the historic (2019) and contemporary (2020, 2021, 2022) comparisons included in our analysis. There was no comparison between the COVID-19 hospitalised cohorts and pneumonia cohort due to the potential for misclassification between COVID-19 and pneumonia.

GP if they determine that the patient's health affects their fitness for work. Unemployed patients can also receive a sick note to support claims for health-related benefits. Sick notes can initially be issued for up to 3 months and then periodically reviewed if needed. During this time, people can receive statutory sick pay.[11]

We identified the first recorded sick note over patient follow-up. Sick notes were identified using Clinical Terms Version 3 codes. Sick notes issued due to COVID-19 did not differ from sick notes issues at other times or for other health conditions. People who were required to isolate beyond 7 days due to having or living with someone with symptoms of COVID-19 would be issued an isolation note, rather than a sick note. Isolation notes did not require contact with a GP and are not counted. Follow-up was censored at the end of the study period (30 November), death or deregistration from their GP practice. The comparator cohorts were also censored if they had a recorded positive SARS-CoV-2 test or COVID-19 diagnosis.

We characterised the demographic and clinical characteristics of patients who received a sick note. Measured demographic variables included age (18–24, 25–34, 35–44, 45–54 and 55–64 years), sex (male, female), GP practice region (East Midlands, East, London, North East, North West, South East, South West, West Midlands and Yorkshire and The Humber), ethnicity (white, Asian and British Asian, black, other and unknown) and IMD quintiles.

Clinical variables included indicators for pre-existing conditions that may impact the severity of COVID-19, specifically: asthma, cancer (haematological, lung, other), chronic cardiac disease, chronic liver disease, chronic respiratory disease (excluding asthma), diabetes, asplenia, HIV infection, hypertension, obesity, organ transplant, neurological conditions, other permanent immunodeficiency (excluding HIV), smoking status (current, former or never) and autoimmune conditions (rheumatoid arthritis, systemic lupus erythematosus, psoriasis). Obesity and smoking status were identified using the most recent recorded information while the remainder were identified at any time prior to the index date.

## Statistical methods

For each of the 11 cohorts, we reported the first sick note rate per 100 person months overall and stratified by demographics (age, sex, ethnicity, region, IMD quintile). To prevent disclosure, all counts in this manuscript are rounded to the nearest 7.

We used Cox regression models to estimate HRs and 95% CIs comparing the first sick note rate between the COVID-19 cohorts and the comparator cohorts. We investigated crude univariable and two covariate-adjusted models: (1) adjusted for age (cubic splines with four knots) and sex and (2) adjusted for demographics and all clinical characteristics described above. While the same people can contribute person-time to multiple exposure groups, these periods are non-overlapping so we applied robust SEs. To determine if the relative differences in sick note occurrence changed over time, we estimated crude and adjusted HRs and 95% CIs both over the entire follow-up period (up to 10 months) and censoring follow-up at 30, 90 and 150 days and examined whether HRs changed according to the duration of follow-up. We chose this approach instead of estimating period-specific HRs in order to avoid selection bias due to the depletion of susceptibles.[12] Similarly, to explore the impact within demographic categories, we estimated crude and adjusted HRs and 95% CIs stratified by age group, sex, ethnicity, IMD quintile and region.

## Software and reproducibility

Data management was performed using Python V.3.8, with analyses carried out using Stata V.16.1, R V.4.3.0 and Python. Code for data management and analysis as well as all codelists used in this study is available online: https://github.com/opensafely/long-covid-sick-notes.

## Patient and public involvement

We have involved patients and the public in various ways: we developed a public website that provides a detailed description of the platform in language suitable for a lay audience (https://opensafely.org); we have participated in two citizen juries exploring public trust in Open-SAFELY; we codeveloped an explainer video (https://www.opensafely.org/about/); we have patient representation who are experts by experience on our OpenSAFELY

Oversight Board; we have partnered with Understanding Patient Data to produce lay explainers on the importance of large datasets for research; we have presented at various online public engagement events to key communities and more. To ensure the patient's voice is represented, we are working closely to decide on language choices with appropriate medical research charities. We will share information and interpretation of our findings through press releases, social media channels, and plain language summaries.

## RESULTS
### Study population
We identified 365 421 people with a recorded positive SARS-CoV-2 test or COVID-19 diagnosis in 2020, of which 22 015 (6.0%) were hospitalised; 1 206 555 people in 2021 (30 205 (2.5%) hospitalised); and 1 321 313 people in 2022 (34 692 (2.6%) hospitalised) (table 1). For comparison with COVID-19 patients who were hospitalised, we identified 29 673 patients hospitalised with pneumonia in 2019. Most people in the COVID-19 cohorts were identified via SGSS testing (88.8% in 2020, 97.8% in 2021 and 96.5% in 2022) (online supplemental figure 1). Compared with earlier years, the 2022 COVID-19 cohort was older and more likely to be female, white ethnicity and in less deprived IMD quintiles (table 1, online supplemental table 1). Hospitalised COVID-19 cohorts tended to be older and were more likely to be male, non-white ethnicity, and in the most deprived IMD quintile compared with the overall COVID-19 cohorts. When compared with the pneumonia cohort, hospitalised COVID-19 patients were more likely to be obese, less likely to be current smokers and had lower rates of many chronic health conditions (online supplemental table 2).

### Overall sick note rates
A total of 34 377 (9.4%), 102 949 (8.5%) and 152 859 (11.6%) people in the COVID-19 cohorts were issued sick notes during follow-up in 2020, 2021 and 2022 (online supplemental table 3). However, after taking into account differences in follow-up time (online supplemental figure 2), the sick note rate among the COVID-19 cohorts decreased over time, from 4.88 per 100 person-months in 2020 (95% CI 4.83 to 4.93) to 2.66 (95% CI 2.64 to 2.67) in 2021 and 1.73 (95% CI 1.72 to 1.73) in 2022 (table 2). The sick note rate was higher in the hospitalised cohorts: 6.78 per 100 person-months (95% CI 6.59 to 6.98) in 2020, 7.19 (95% CI 7.03 to 7.36) in 2021 and 4.13 (95% CI 4.03 to 4.22) in 2022. In contrast, the sick note rate in the pneumonia cohort was 7.17 (95% CI 7.01 to 7.34) (online supplemental table 4).

### Sick note rate by demographics
Among people in the COVID-19 cohorts, the sick note rate was higher in 2020 and lower in 2022 for all demographic groups except people 18–24 years (table 2). Generally, the sick note rate was higher in people aged ≥45 years, women, people of Asian or Asian British ethnicity, and lower in people 18–24 years and living in London. The sick note rate increased with greater deprivation as defined by the IMD quintile. In most cases, these patterns reflected those in the general population. A slightly different pattern was observed for the hospitalised cohorts (online supplemental table 4). People 45–54 years were most likely to receive a sick note, which was older than the pneumonia cohort. Among hospitalised patients, the relationship between sex and receiving a sick note differed by year, with women more likely to receive a sick note in 2020 and men more likely in 2021 and 2022. A linear relationship between lesser deprivation and a higher sick note rate was seen for the pneumonia cohort but not the hospitalised COVID-19 cohorts.

### Cox regression by follow-up period
Most people in the COVID-19 cohorts who were issued a first sick note received it in the first 30 days postindex date (table 3). The fully adjusted HR representing the average effect over the entire study period was highest comparing the 2020 COVID-19 cohort to the 2020 general population (4.07, 95% CI 4.02 to 4.12) and lowest comparing the 2022 COVID-19 cohort to the 2022 general population (1.57, 95% CI 1.56 to 1.58) (table 3). The crude and age-sex-adjusted HRs are in online supplemental table 5. For all comparisons, the HR was greatest in the first 30 days from the index date, with fully adjusted HRs ranging from 5.94 (95% CI 5.85 to 6.03) comparing the 2020 COVID-19 cohort to the 2020 general population to 2.37 (95% CI 2.34 to 2.39) comparing the 2022 COVID-19 cohort to the 2022 general population. With longer follow-up periods, the HR attenuated but remained high. For the hospitalised cohorts, the overall sick note rate was lower in the COVID-19 cohorts than the pneumonia cohort with the highest rates in the first 30 days. The average HR overall follow-up time ranged from 0.62 (95% CI 0.59 to 0.65) for the 2022 COVID-19 cohort to 0.81 (95% CI 0.77 to 0.86) for the 2021 cohort (online supplemental table 6). The HR was relatively stable regardless of follow-up period used.

### Cox regression stratified by demographic categories
For all demographic groups, the HR was greatest in 2020 when compared with either a contemporary or 2019 historical comparator (figure 2, online supplemental tables 7–11). In 2020, large relative increases in sick note rates were observed for people ≥35 years, women and people of black or other ethnicity. The HR was also higher in people in less deprived IMD quintiles when comparing the 2020 COVID-19 cohort to the 2020 general population, the fully adjusted HR ranged from 3.41 (95% CI 3.33 to 3.48) in the most deprived quintile to 5.15 (95% CI 4.98 to 5.32) in the least deprived quintile. Most HRs were lower in 2021 compared with 2020, and more so in 2022. The one exception was people 18–24 years, where the HR compared with the general population was 1.47 (95% CI 1.43 to 1.51) in 2022, compared with 1.24 (95%

**Table 1** Demographic and select clinical characteristics of general population COVID-19 and comparator cohorts, 1 February–30 November of each year

| | 2019 | 2020 | | 2021 | | 2022 | |
|---|---|---|---|---|---|---|---|
| | General population* | COVID-19 cohort | General population† | COVID-19 cohort | General population† | COVID-19 cohort | General population† |
| | n (%) | n (%) | n (%) | n (%) | n (%) | n (%) | n (%) |
| Total | 3 140 326 (100.0) | 365 421 (100.0) | 3 439 534 (100.0) | 1 206 555 (100.0) | 4 571 469 (100.0) | 1 321 313 (100.0) | 4 818 870 (100.0) |
| Age | | | | | | | |
| 18–24 years | 496825 (15.8) | 61355 (16.8) | 545286 (15.9) | 196616 (16.3) | 699993 (15.3) | 111342 (8.4) | 375704 (7.8) |
| 25–34 years | 663495 (21.1) | 77686 (21.3) | 726712 (21.1) | 264159 (21.9) | 989506 (21.6) | 258314 (19.5) | 915152 (19.0) |
| 35–44 years | 644945 (20.5) | 74767 (20.5) | 706692 (20.5) | 294238 (24.4) | 1116283 (24.4) | 312732 (23.7) | 1142764 (23.7) |
| 45–54 years | 719663 (22.9) | 83832 (22.9) | 787465 (22.9) | 271418 (22.5) | 1042601 (22.8) | 328510 (24.9) | 1197693 (24.9) |
| 55–64 years | 615405 (19.6) | 67781 (18.5) | 673372 (19.6) | 180117 (14.9) | 723093 (15.8) | 310422 (23.5) | 1187557 (24.6) |
| Sex | | | | | | | |
| Female | 1732654 (55.2) | 201824 (55.2) | 1898351 (55.2) | 645666 (53.5) | 2447200 (53.5) | 823438 (62.3) | 3001922 (62.3) |
| Male | 1407672 (44.8) | 163597 (44.8) | 1541183 (44.8) | 560889 (46.5) | 2124269 (46.5) | 497875 (37.7) | 1816948 (37.7) |
| Ethnicity | | | | | | | |
| White | 1914458 (61.0) | 213185 (58.3) | 2109478 (61.3) | 788249 (65.3) | 2866892 (62.7) | 967225 (73.2) | 3287305 (68.2) |
| Asian or Asian British | 225316 (7.2) | 45101 (12.3) | 254884 (7.4) | 68915 (5.7) | 315854 (6.9) | 52101 (3.9) | 300167 (6.2) |
| Black | 67319 (2.1) | 8099 (2.2) | 76622 (2.2) | 21399 (1.8) | 103117 (2.3) | 18340 (1.4) | 105322 (2.2) |
| Mixed | 34748 (1.1) | 4298 (1.2) | 39872 (1.2) | 13937 (1.2) | 56644 (1.2) | 14077 (1.1) | 58527 (1.2) |
| Other | 60935 (1.9) | 5502 (1.5) | 70427 (2.0) | 14910 (1.2) | 102585 (2.2) | 19425 (1.5) | 105007 (2.2) |
| Unknown | 837550 (26.7) | 89236 (24.4) | 888265 (25.8) | 299145 (24.8) | 1126384 (24.6) | 250152 (18.9) | 962542 (20.0) |
| IMD | | | | | | | |
| 1 (most deprived) | 795739 (25.3) | 98399 (26.9) | 877583 (25.5) | 257992 (21.4) | 1027467 (22.5) | 212044 (16.0) | 929649 (19.3) |
| 2 | 630189 (20.1) | 76874 (21.0) | 691131 (20.1) | 243334 (20.2) | 926135 (20.3) | 251692 (19.0) | 968303 (20.1) |
| 3 | 605759 (19.3) | 68887 (18.9) | 661150 (19.2) | 248696 (20.6) | 935067 (20.5) | 292131 (22.1) | 1047914 (21.7) |
| 4 | 57,9614 (18.5) | 64064 (17.5) | 633059 (18.4) | 235473 (19.5) | 877170 (19.2) | 286538 (21.7) | 975576 (20.2) |
| 5 (least deprived) | 52,9025 (16.8) | 57190 (15.7) | 576618 (16.8) | 221067 (18.3) | 805623 (17.6) | 278915 (21.1) | 897421 (18.6) |
| Region | | | | | | | |
| East | 432551 (13.8) | 50246 (13.8) | 473928 (13.8) | 240170 (19.9) | 910567 (19.9) | 327096 (24.8) | 1184981 (24.6) |
| East Midlands | 645218 (20.5) | 75040 (20.5) | 706370 (20.5) | 227577 (18.9) | 863884 (18.9) | 222397 (16.8) | 814919 (16.9) |
| London | 145264 (4.6) | 17073 (4.7) | 158844 (4.6) | 56693 (4.7) | 210623 (4.6) | 71113 (5.4) | 255997 (5.3) |
| North East | 257985 (8.2) | 29988 (8.2) | 282198 (8.2) | 71169 (5.9) | 270284 (5.9) | 59059 (4.5) | 216216 (4.5) |
| North West | 426615 (13.6) | 49623 (13.6) | 467383 (13.6) | 129934 (10.8) | 494144 (10.8) | 108857 (8.2) | 399168 (8.3) |

Continued

**Table 1** Continued

| | 2019 | 2020 | | 2021 | | 2022 | |
| | General population* | COVID-19 cohort | General population† | COVID-19 cohort | General population† | COVID-19 cohort | General population† |
| | n (%) | n (%) | n (%) | n (%) | n (%) | n (%) | n (%) |
|---|---|---|---|---|---|---|---|
| South East | 101 976 (3.2) | 11 942 (3.3) | 111 986 (3.3) | 70 763 (5.9) | 266 098 (5.8) | 100 114 (7.6) | 363 867 (7.6) |
| South West | 220 759 (7.0) | 25 641 (7.0) | 241 500 (7.0) | 160 678 (13.3) | 608 790 (13.3) | 227 731 (17.2) | 833 742 (17.3) |
| West Midlands | 175 581 (5.6) | 20 559 (5.6) | 192 836 (5.6) | 51 737 (4.3) | 194 593 (4.3) | 40 880 (3.1) | 147 791 (3.1) |
| Yorkshire and The Humber | 734 370 (23.4) | 85 309 (23.3) | 804 496 (23.4) | 197 827 (16.4) | 752 486 (16.5) | 164 073 (12.4) | 602 175 (12.5) |
| Common health conditions | | | | | | | |
| Obesity | 681 422 (21.7) | 96 712 (26.5) | 748 398 (21.8) | 276 759 (22.9) | 957 194 (20.9) | 350 777 (26.5) | 1 137 710 (23.6) |
| Hypertension | 325 969 (10.4) | 43 610 (11.9) | 353 416 (10.3) | 112 147 (9.3) | 411 803 (9.0) | 164 367 (12.4) | 564 823 (11.7) |
| Diabetes | 184 450 (5.9) | 30 590 (8.4) | 210 532 (6.1) | 70 861 (5.9) | 262 906 (5.8) | 100 541 (7.6) | 358 449 (7.4) |
| Asthma | 538 657 (17.2) | 68 208 (18.7) | 595 763 (17.3) | 236 439 (19.6) | 807 618 (17.7) | 279 944 (21.2) | 852 586 (17.7) |

General population comparator cohorts were frequency matched in age, sex and administrative region. Additional health conditions are presented in online supplemental table 1. All counts rounded to nearest 7.
*Age, sex and STP frequency matched with 2020 COVID-19 cohort.
†Age, sex and STP frequency matched with contemporary COVID-19 cohort.
STP, sustainability and transformation partnership.

**Table 2** Rate of first sick note per 100 person-months for each cohort by demographics

| | 2019 | 2020 | | 2021 | | 2022 | |
| --- | --- | --- | --- | --- | --- | --- | --- |
| | General population* Rate per 100 person-months (95% CI) | COVID-19 cohort Rate per 100 person-months (95% CI) | General population* Rate per 100 person-months (95% CI) | COVID-19 cohort Rate per 100 person-months (95% CI) | General population* Rate per 100 person-months (95% CI) | COVID-19 cohort Rate per 100 person-months (95% CI) | General population* Rate per 100 person-months (95% CI) |
| Total | 1.21 (1.21 to 1.21) | 4.88 (4.83 to 4.93) | 0.93 (0.93 to 0.93) | 2.66 (2.64 to 2.67) | 1.23 (1.23 to 1.23) | 1.73 (1.72 to 1.73) | 1.23 (1.23 to 1.23) |
| **Age** | | | | | | | |
| 18–24 years | 0.98 (0.97 to 0.99) | 1.58 (1.50 to 1.65) | 0.66 (0.65 to 0.66) | 1.25 (1.23 to 1.28) | 0.96 (0.95 to 0.98) | 1.30 (1.28 to 1.33) | 0.91 (0.90 to 0.92) |
| 25–34 years | 1.18 (1.16 to 1.19) | 3.51 (3.42 to 3.61) | 0.86 (0.85 to 0.87) | 2.16 (2.13 to 2.19) | 1.17 (1.16 to 1.18) | 1.62 (1.60 to 1.64) | 1.14 (1.13 to 1.15) |
| 35–44 years | 1.19 (1.18 to 1.20) | 5.19 (5.07 to 5.31) | 0.91 (0.90 to 0.92) | 2.95 (2.92 to 2.99) | 1.24 (1.23 to 1.25) | 1.65 (1.63 to 1.66) | 1.19 (1.18 to 1.20) |
| 45–54 years | 1.35 (1.34 to 1.37) | 6.53 (6.40 to 6.65) | 1.08 (1.07 to 1.09) | 3.60 (3.55 to 3.64) | 1.38 (1.37 to 1.39) | 1.91 (1.90 to 1.93) | 1.35 (1.34 to 1.36) |
| 55–64 years | 1.30 (1.28 to 1.31) | 6.62 (6.48 to 6.75) | 1.09 (1.07 to 1.10) | 3.79 (3.74 to 3.85) | 1.33 (1.32 to 1.35) | 1.87 (1.85 to 1.89) | 1.32 (1.32 to 1.33) |
| **Sex** | | | | | | | |
| Female | 1.39 (1.38 to 1.40) | 5.88 (5.80 to 5.95) | 1.11 (1.10 to 1.11) | 3.25 (3.22 to 3.27) | 1.46 (1.46 to 1.47) | 1.96 (1.95 to 1.97) | 1.41 (1.41 to 1.42) |
| Male | 0.99 (0.98 to 1.00) | 3.61 (3.54 to 3.68) | 0.72 (0.71 to 0.72) | 2.03 (2.01 to 2.05) | 0.96 (0.95 to 0.97) | 1.35 (1.34 to 1.37) | 0.94 (0.93 to 0.94) |
| **Ethnicity** | | | | | | | |
| White | 1.30 (1.30 to 1.31) | 5.17 (5.10 to 5.24) | 1.00 (0.99 to 1.00) | 2.79 (2.77 to 2.81) | 1.31 (1.31 to 1.32) | 1.73 (1.72 to 1.74) | 1.29 (1.28 to 1.29) |
| Mixed | 1.03 (1.01 to 1.04) | 4.83 (4.69 to 4.97) | 0.86 (0.84 to 0.87) | 3.11 (3.05 to 3.18) | 1.10 (1.08 to 1.12) | 2.07 (2.02 to 2.12) | 1.16 (1.14 to 1.18) |
| Asian/Asian British | 1.24 (1.20 to 1.28) | 5.51 (5.18 to 5.84) | 0.91 (0.88 to 0.94) | 3.47 (3.34 to 3.61) | 1.29 (1.25 to 1.32) | 2.22 (2.14 to 2.31) | 1.42 (1.38 to 1.45) |
| Black | 1.22 (1.16 to 1.27) | 4.55 (4.11 to 4.98) | 0.92 (0.88 to 0.96) | 2.65 (2.51 to 2.80) | 1.19 (1.15 to 1.23) | 1.74 (1.66 to 1.83) | 1.25 (1.21 to 1.29) |
| Other | 0.62 (0.59 to 0.65) | 3.91 (3.56 to 4.25) | 0.47 (0.44 to 0.49) | 2.31 (2.18 to 2.44) | 0.66 (0.64 to 0.69) | 1.30 (1.24 to 1.37) | 0.75 (0.72 to 0.77) |
| Unknown | 1.09 (1.08 to 1.10) | 4.23 (4.13 to 4.33) | 0.83 (0.82 to 0.84) | 2.19 (2.16 to 2.22) | 1.10 (1.09 to 1.11) | 1.62 (1.60 to 1.64) | 1.09 (1.08 to 1.10) |
| **IMD quintile** | | | | | | | |
| 1 (most deprived) | 1.58 (1.57 to 1.59) | 5.27 (5.17 to 5.38) | 1.20 (1.19 to 1.21) | 3.13 (3.10 to 3.17) | 1.66 (1.65 to 1.67) | 2.45 (2.42 to 2.48) | 1.71 (1.69 to 1.72) |
| 2 | 1.33 (1.32 to 1.34) | 5.07 (4.96 to 5.18) | 1.02 (1.01 to 1.03) | 2.84 (2.80 to 2.88) | 1.35 (1.34 to 1.36) | 2.01 (1.99 to 2.03) | 1.38 (1.37 to 1.39) |
| 3 | 1.12 (1.11 to 1.13) | 4.84 (4.72 to 4.95) | 0.87 (0.86 to 0.88) | 2.57 (2.54 to 2.61) | 1.13 (1.12 to 1.14) | 1.65 (1.64 to 1.67) | 1.15 (1.14 to 1.16) |
| 4 | 1.00 (0.99 to 1.01) | 4.65 (4.53 to 4.77) | 0.77 (0.76 to 0.78) | 2.38 (2.34 to 2.41) | 1.02 (1.01 to 1.03) | 1.50 (1.48 to 1.52) | 1.03 (1.02 to 1.04) |
| 5 (least deprived) | 0.86 (0.85 to 0.88) | 4.23 (4.11 to 4.35) | 0.66 (0.65 to 0.67) | 2.18 (2.14 to 2.22) | 0.88 (0.87 to 0.89) | 1.27 (1.26 to 1.29) | 0.90 (0.89 to 0.91) |
| **Region** | | | | | | | |
| East | 1.21 (1.19 to 1.22) | 4.79 (4.67 to 4.91) | 0.92 (0.91 to 0.93) | 2.74 (2.71 to 2.78) | 1.29 (1.28 to 1.30) | 1.89 (1.87 to 1.91) | 1.34 (1.33 to 1.35) |
| East Midlands | 1.05 (1.03 to 1.06) | 4.06 (3.94 to 4.17) | 0.76 (0.75 to 0.77) | 2.39 (2.36 to 2.43) | 1.05 (1.04 to 1.05) | 1.53 (1.51 to 1.54) | 1.08 (1.07 to 1.09) |
| London | 0.71 (0.69 to 0.73) | 3.08 (2.90 to 3.26) | 0.49 (0.48 to 0.51) | 2.07 (2.01 to 2.14) | 0.71 (0.69 to 0.73) | 1.22 (1.19 to 1.25) | 0.76 (0.74 to 0.77) |
| North East | 1.34 (1.32 to 1.36) | 5.32 (5.13 to 5.51) | 1.06 (1.05 to 1.08) | 2.88 (2.81 to 2.95) | 1.43 (1.41 to 1.45) | 2.09 (2.04 to 2.14) | 1.52 (1.50 to 1.55) |

**Table 2** Continued

| | 2019 | 2020 | | 2021 | | 2022 | |
| | General population* Rate per 100 person-months (95% CI) | COVID-19 cohort Rate per 100 person-months (95% CI) | General population* Rate per 100 person-months (95% CI) | COVID-19 cohort Rate per 100 person-months (95% CI) | General population* Rate per 100 person-months (95% CI) | COVID-19 cohort Rate per 100 person-months (95% CI) | General population* Rate per 100 person-months (95% CI) |
|---|---|---|---|---|---|---|---|
| North West | 1.39 (1.38 to 1.41) | 5.76 (5.61 to 5.92) | 1.12 (1.11 to 1.14) | 2.81 (2.76 to 2.86) | 1.51 (1.50 to 1.53) | 2.11 (2.08 to 2.14) | 1.53 (1.51 to 1.54) |
| South East | 0.96 (0.93 to 0.98) | 3.73 (3.50 to 3.96) | 0.72 (0.70 to 0.74) | 2.24 (2.17 to 2.30) | 1.00 (0.98 to 1.02) | 1.44 (1.41 to 1.47) | 1.03 (1.02 to 1.05) |
| South West | 1.07 (1.05 to 1.09) | 4.55 (4.37 to 4.74) | 0.81 (0.79 to 0.82) | 2.57 (2.52 to 2.62) | 1.12 (1.11 to 1.13) | 1.61 (1.59 to 1.63) | 1.15 (1.14 to 1.16) |
| West Midlands | 1.32 (1.29 to 1.34) | 5.63 (5.38 to 5.88) | 0.97 (0.95 to 0.99) | 3.00 (2.92 to 3.07) | 1.37 (1.34 to 1.39) | 2.20 (2.15 to 2.26) | 1.49 (1.46 to 1.52) |
| Yorkshire and The Humber | 1.31 (1.30 to 1.33) | 5.46 (5.34 to 5.58) | 1.03 (1.02 to 1.04) | 2.93 (2.89 to 2.97) | 1.41 (1.39 to 1.42) | 2.01 (1.98 to 2.03) | 1.44 (1.43 to 1.45) |

*Age, sex and STP frequency matched with 2020 COVID-19 cohort.

CI 1.22 to 1.27) in 2021. Other patterns that differed in 2022 compared with other years were that no meaningful variation by IMD quintile was observed, and a higher HR was seen in men compared with women. The findings were similar comparing the COVID-19 cohorts with the 2019 comparator. Among the hospitalised cohorts, the HRs comparing them with the pneumonia cohort were similar in 2020 and 2021, but lower in 2022 (online supplemental figure 3 and tables 7–11). A lower sick note rate in the COVID-19 cohorts compared with the pneumonia cohort was observed for most demographic subgroups.

## DISCUSSION
### Summary
We found that people with a recorded positive SARS-CoV-2 test or COVID-19 diagnosis had a higher sick note rate than the general population, even after adjusting for demographics and a wide range of clinical characteristics. This increase was greatest in 2020 but continued even into 2022 despite the introduction of vaccines and other effective outpatient COVID-19 treatments. The sick note rate was highest in the first 30 days postdiagnosis, but an increased risk persisted overall follow-up in all years. These findings provide further evidence for the long-term health and economic impact of COVID-19. In contrast, people hospitalised with COVID-19 were less likely to be issued a sick note than people with pneumonia, especially in 2022.

### Strengths and weaknesses
Our data come from a representative sample of 40% of the English population.[13] To better understand sick note patterns, we used COVID-19 cohorts from three different pandemic time periods, with both contemporary and historical comparators, to account for changes in preventive measures (eg, lockdowns, vaccines) and other disruptions to working patterns (eg, furlough) that would have impacted on receipt of sick notes over time. However, there is bias in who gets tested or seeks care for COVID-19 which will have changed as the pandemic evolved,[14] especially after the cessation of free testing in April 2022. It is unclear how this would correlate with the likelihood of requesting or requiring a sick note. To mitigate this, we relied on three different methods for identifying COVID-19 cases (positive PCR or LFT test, primary care diagnosis, hospitalisation), but many mild or asymptomatic cases, and cases among people choosing not to get tested or not to record their LFT result will be missed.

We also could not identify whether people were participating in the workforce and included everyone of working age. From 2019 to 2022, the employment rate was 62% in people 18–24 years, 85% in people 25–49 years and 72% in people 50–64 years.[15] It is, therefore, likely that the true absolute sick note rate will be substantially attenuated, especially among the younger age groups. However, this will only impact on the relative effects if the employment

**Table 3** Adjusted HRs for receipt of first sick note by follow-up period postindex date

| Comparison and follow-up period postindex date | Sick note rate per 100 person-months (95% CI) | | Adjusted HR* (95% CI) |
|---|---|---|---|
| | COVID-19 cohort | General population† | |
| **2020 COVID-19 cohort vs 2019 general population** | | | |
| 0–30 days | 9.35 (9.24 to 9.46) | 1.99 (1.98 to 2.01) | 4.68 (4.61 to 4.75) |
| 0–90 days | 6.20 (6.14 to 6.27) | 1.48 (1.47 to 1.48) | 3.63 (3.58 to 3.68) |
| 0–150 days | 5.39 (5.33 to 5.45) | 1.31 (1.30 to 1.32) | 3.36 (3.32 to 3.40) |
| Entire follow-up period | 4.88 (4.83 to 4.93) | 1.21 (1.20 to 1.22) | 3.19 (3.15 to 3.23) |
| **2020 COVID-19 cohort vs 2020 general population** | | | |
| 0–30 days | 9.35 (9.24 to 9.46) | 1.57 (1.55 to 1.58) | 5.94 (5.85 to 6.03) |
| 0–90 days | 6.20 (6.14 to 6.27) | 1.12 (1.11 to 1.13) | 4.67 (4.61 to 4.73) |
| 0–150 days | 5.39 (5.33 to 5.45) | 0.99 (0.99 to 1.00) | 4.32 (4.27 to 4.38) |
| Entire follow-up period | 4.88 (4.83 to 4.93) | 0.93 (0.93 to 0.93) | 4.07 (4.02 to 4.12) |
| **2021 COVID-19 cohort vs 2019 general population** | | | |
| 0–30 days | 6.51 (6.46 to 6.56) | 1.99 (1.98 to 2.01) | 3.34 (3.30 to 3.38) |
| 0–90 days | 3.50 (3.48 to 3.52) | 1.48 (1.47 to 1.48) | 2.38 (2.35 to 2.41) |
| 0–150 days | 2.91 (2.90 to 2.93) | 1.31 (1.30 to 1.32) | 2.15 (2.13 to 2.17) |
| Entire follow-up period | 2.66 (2.64 to 2.67) | 1.21 (1.20 to 1.22) | 2.02 (2.00 to 2.04) |
| **2021 COVID-19 cohort vs 2021 general population** | | | |
| 0–30 days | 6.51 (6.46 to 6.56) | 1.97 (1.96 to 1.99) | 3.37 (3.33 to 3.40) |
| 0–90 days | 3.50 (3.48 to 3.52) | 1.47 (1.46 to 1.48) | 2.35 (2.33 to 2.37) |
| 0–150 days | 2.91 (2.90 to 2.93) | 1.32 (1.31 to 1.33) | 2.12 (2.10 to 2.13) |
| Entire follow-up period | 2.66 (2.64 to 2.67) | 1.23 (1.22 to 1.23) | 1.99 (1.98 to 2.01) |
| **2022 COVID-19 cohort vs 2019 general population** | | | |
| 0–30 days | 4.87 (4.83 to 4.91) | 1.99 (1.98 to 2.01) | 2.27 (2.24 to 2.30) |
| 0–90 days | 2.57 (2.55 to 2.58) | 1.48 (1.47 to 1.48) | 1.73 (1.72 to 1.75) |
| 0–150 days | 2.05 (2.04 to 2.06) | 1.31 (1.30 to 1.32) | 1.63 (1.61 to 1.64) |
| Entire follow-up period | 1.73 (1.72 to 1.73) | 1.21 (1.20 to 1.22) | 1.55 (1.54 to 1.57) |
| **2022 COVID-19 cohort vs 2022 general population** | | | |
| 0–30 days | 4.87 (4.83 to 4.91) | 2.09 (2.07 to 2.10) | 2.37 (2.34 to 2.39) |
| 0–90 days | 2.57 (2.55 to 2.58) | 1.53 (1.52 to 1.54) | 1.73 (1.73 to 1.76) |
| 0–150 days | 2.05 (2.04 to 2.06) | 1.34 (1.39 to 1.35) | 1.64 (1.62 to 1.65) |
| Entire follow-up period | 1.73 (1.72 to 1.73) | 1.23 (1.22 to 1.23) | 1.57 (1.56 to 1.58) |

Entire follow-up period was up to 10 months. Crude and age-sex-adjusted HRs presented in online supplemental table 6.
*Fully adjusted models include age, sex, IMD quintile, region, ethnicity, obesity, smoking status, hypertension, diabetes, chronic respiratory disease, asthma, chronic cardiac disease, lung cancer, haematological cancer, other cancer, chronic liver disease, other neurological disease, organ transplant, asplenia, HIV, permanent immunodeficiency and rheumatoid arthritis/systemic lupus erythematosus/psoriasis.
†General population age, sex and STP frequency matched with COVID-19 cohort.
IMD, index of multiple deprivation; STP, sustainability and transformation partnership.

rate differs substantially between the COVID-19 cohorts and the general population. Last, people with very severe symptoms who need to leave the workforce completely may also not be captured in our data, but these counts are likely to be small.

### Findings in context
The introduction of vaccination programmes and differences in circulating variants lead to changes in severity and transmissibility of SARS-CoV-2. While the infection rate went up during the study period, the COVID-19-related mortality rate has decreased in each subsequent wave, in large part due to greater vaccination coverage.[7] This reduced severity helps explain the lower sick note rate in later years. Additionally, the ending of lockdowns and other preventive measures, cessation of free testing and differential uptake of vaccines by demographic and clinical subgroups will have impacted on the characteristics of people testing positive for SARS-CoV-2.[16] Consistent

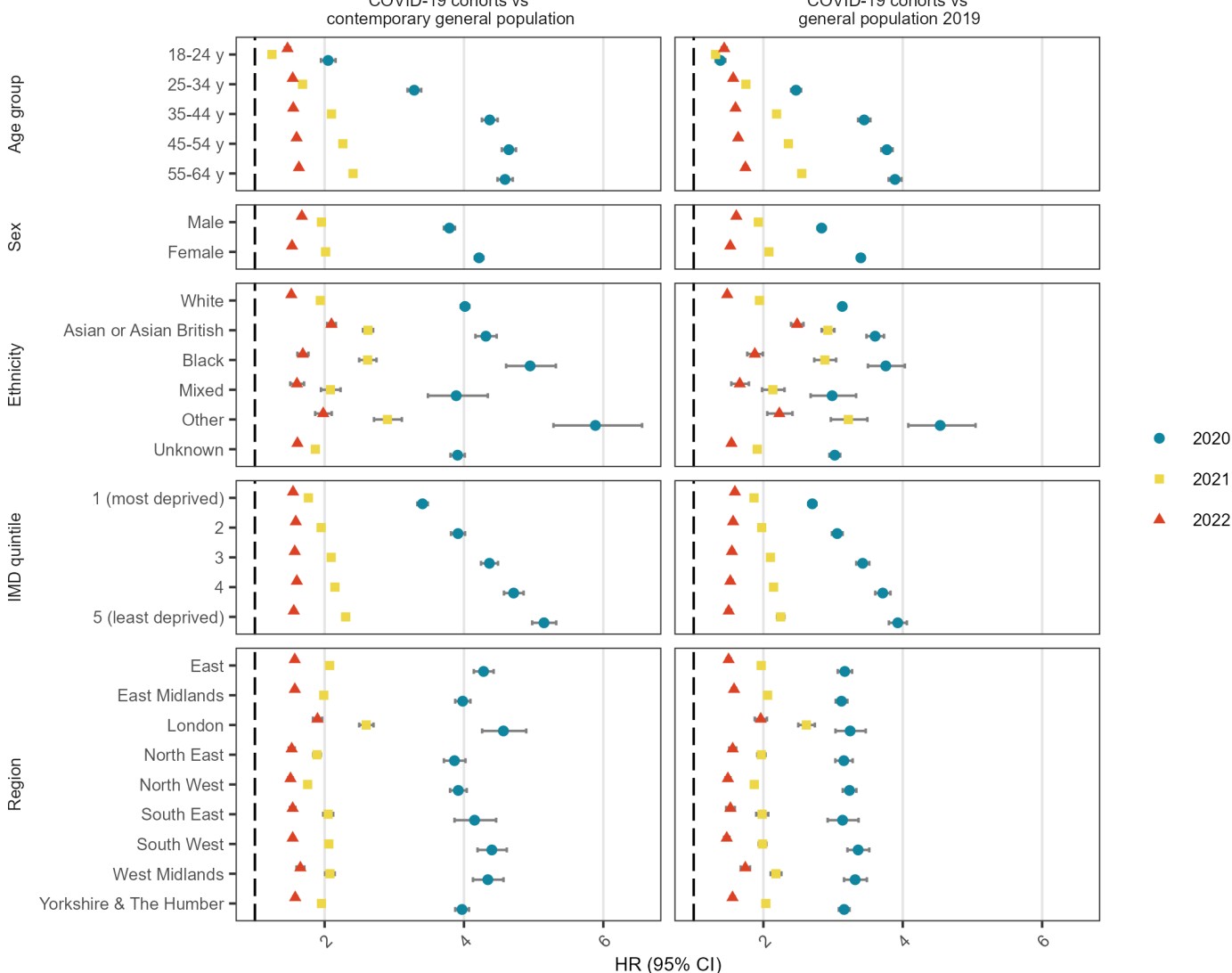

**Figure 2** Fully adjusted HR of first sick note comparing COVID-19 cohorts to their contemporary and historical 2019 general population, stratified by demographic categories. All models are adjusted for age, sex, ethnicity, IMD quintile and region but exclude the stratification variable. Models are additionally adjusted for obesity, smoking status, hypertension, diabetes, chronic respiratory disease, asthma, chronic cardiac disease, lung cancer, haematological cancer, other cancer, chronic liver disease, other neurological disease, organ transplant, asplenia, HIV, permanent immunodeficiency and rheumatoid arthritis/systemic lupus erythematosus/psoriasis. IMD, index of multiple deprivation.

with previously reported data,[17] we observed that the number of people with evidence of a positive SARS-CoV-2 test was much greater in 2021 and 2022 compared with 2020. Thus, while the sick note rate was nearly twice as high in 2020, the actual number of people issued a sick note was 4.5 times higher in 2022. Although people hospitalised with pneumonia were older and had more comorbidities, after adjustment for demographics and clinical characteristics people hospitalised with COVID-19 were still less likely to be issued a sick note than people with pneumonia, especially in 2022 suggesting the long-term effects are not worse than comparable serious respiratory infections requiring hospitalisation.

In contrast, the sick note rate was lower in the general population in 2020, compared with other years. This is consistent with a previous study of NHS workers which

identified a decrease in sick notes for non-COVID-related episodes in 2020.[18] There are likely to be several explanations such as changes to working patterns, including furlough and remote working, which would have reduced the need for sick notes. There were also fewer circulating non-COVID-19 respiratory infections in 2020 due to public health measures.[19] However, a previous study identified that while the sick note rate decreased overall in 2020, sick notes associated with certain diagnoses, specifically asthma, respiratory conditions and mental health increased during the COVID-19 pandemic.[18] Similarly, a study of NHS workers found that sickness absences due to psychiatric illness were higher in 2022 compared with prepandemic.[20]

Direct comparisons with studies of sickness absences in other jurisdictions are difficult due to differences in

study populations, follow-up periods and how sick leave is defined. In a large Swedish study (n=661 780), 6% experienced long-term sick leave associated with COVID-19, more common in women, older people and people with lower incomes, more comorbidities or hospitalised with COVID-19.[21] Other studies focused either on people with COVID-19 only or certain populations. In Wales, 15% of domiciliary care workers (n=15 931) were issued a sick note between March 2020 and November 2021, and they were more common in women and older people.[22] In Germany, a study of 30 950 people with COVID-19 found that 6% experienced long-term sick leave.[23]

## Policy implications and future research

Now several years into the pandemic, vaccines have led to reduced severity of COVID-19, but the number of daily cases remains high,[16] potentially putting many people at risk of long-term symptoms. The incidence of repeat infections is also increasing;[24] while a repeat infection appears to lead to milder acute symptoms,[25 26] other studies have found that the long-term symptom burden increases with the number of reinfections.[27 28] In our study, we did not try and link the reason for the sick note to COVID-19. However, most studies of long-term effects have focused on surveys and self-report, which can be unrepresentative[29–31] and long COVID is not well coded in EHRs.[10] Thus, our findings provide a different perspective on the potential long-term consequences of COVID-19 that does not rely on accurate coding of COVID-19-related symptoms.

In the UK, the sickness absence rate in 2022 is the highest it has been since 2004.[32] The populations with the highest absolute rates of sick notes after COVID-19 (women, people living in areas of greater deprivation) are those over-represented in low-income and/or public-facing jobs, which have higher rates of sickness absences.[33] Mental health problems are both a common cause of long-term sickness absence[1 34] and a predictor of long COVID,[31] pointing to unmet needs in these populations. However, although people living in areas of least deprivation with COVID-19 had the lowest absolute sick note rate, they had the greatest relative increase compared with the general population in earlier years, reflecting their lower baseline rate; this pattern disappeared in 2022.

Pressures on primary care are also increasing, and over the past decade the number of GPs has gone down and each GP is thus responsible for a greater number of patients.[35] It is unclear to what extent the increasing number of people experiencing long-term postacute symptoms contributes to the burden on primary care. Thus, we need to understand not only how people with long COVID symptoms interact with primary care, but how their symptoms are recorded in EHRs to inform future studies. In addition, further investigation into the duration of sick notes, as well as the diagnoses associated with the issued sick note, whether these changed over time from diagnosis, and how these compare to sick notes issued to people without COVID-19 will help shed light on the most common long-term symptoms experienced.

## CONCLUSION

Despite undercapture of people with COVID-19 and over-estimation of the number of people in the workforce, we have identified a considerable increased risk of sick notes among people with COVID-19 compared with the general population. This extends into an era of high vaccination rates and when the severity of illness from COVID-19 is decreasing. The majority of sick notes were issued within the first 30 days postdiagnosis, suggesting that most COVID-19-related sick leave is associated with the acute phase of the disease. However, a persistent increased risk up to 10 months after the illness demonstrates the ongoing health and economic impact of COVID-19.

**Author affiliations**
[1]Bennett Institute for Applied Data Science, Nuffield Department of Primary Care Health Sciences, Unviersity of Oxford, Oxford, UK
[2]London School of Hygiene and Tropical Medicine, London, UK
[3]Institute of Health Informatics, University College London, London, UK
[4]University College London Hospitals Biomedical Research Centre, London, UK
[5]BHF Data Science Centre, Health Data Research UK, London, UK
[6]TPP, Leeds, UK
[7]National Institute for Health and Care Excellence, London, UK
[8]Department of Primary Care and Public Health, Imperial College London, London, UK
[9]NIHR Applied Research Collaboration West, Bristol, UK

**Acknowledgements** We are very grateful for all the support received from the TPP Technical Operations team throughout this work and for generous assistance from the information governance and database teams at NHS England and the NHS England Transformation Directorate.

**Collaborators** Members of OpenSAFELY Collaborative: Sebastian CJ Bacon, Lucy Bridges, Benjamin FC Butler-Cole, Simon Davy, Iain Dillingham, David Evans, Ben Goldacre, Liam Hart, George Hickman, Peter Inglesby, Steven Maude, Jessica Morley, Amir Mehrkar, Thomas O'Dwyer, Rebecca M Smith, Tom Ward, Jon Massey, Milan Wiedemann, Christopher Bates, Jonathan Cockburn, Sam Harper, Frank Hester, John Parry. Members of National Core Studies Collaborative: Nishi Chaturvedi, Chloe Park, Alisia Carnemolla, Dylan Williams, Anika Knueppel, Andy Boyd, Emma L Turner, Katharine M Evans, Richard Thomas, Samantha Berman, Stela McLachlan, Matthew Crane, Rebecca Whitehorn, Jacqui Oakley, Diane Foster, Hannah Woodward, Kirsteen C Campbell, Nicholas Timpson, Alex Kwong, Ana Goncalves Soares, Gareth Griffith, Renin Toms, Louise Jones, Annie Herbert, Ruth Mitchell, Tom Palmer, Jonathan Sterne, Venexia Walker, Lizzie Huntley, Laura Fox, Rachel Denholm, Rochelle Knight, Kate, Northstone, Arun, Kanagaratnam, Elsie Horne, Harriet Forbes, Teri North, Kurt Taylor, Marwa AL Arab, Scott Walker, Jose IC Coronado, Arun S Karthikeyan, George, Ploubidis, Bettina Moltrecht, Charlotte Booth, Sam Parsons, Bozena Wielgoszewska, Charis Bridger-Staatz, Claire Steves, Ellen Thompson, Paz Garcia, Nathan Cheetham, Ruth Bowyer, Maxim Freydin, Amy Roberts, Ben Goldacre, Alex Walker, Jess Morley, William Hulme, Linda Nab, Louis Fisher, Brian MacKenna, Colm Andrews, Helen Curtis, Lisa Hopcroft, Amelia Green, Praveetha Patalay, Jane Maddock, Kishan Patel, Jean Stafford, Wels Jacques, Kate Tilling, John Macleod, Eoin McElroy, Anoop ShahRichard Silverwood, Spiros Denaxas, Robin Flaig, Daniel McCartney, Archie Campbell, Laurie Tomlinson, John Tazare, Bang Zheng, Liam Smeeth, Emily Herrett, Thomas Cowling, Kate Mansfield, Ruth E Costello, Kevin Wang, Kathryn Mansfield, Viyaasan Mahalingasivam, Ian Douglas, Sinead Langan, Sinead Brophy, Michael Parker, Jonathan Kennedy, Rosie McEachan, John Wright, Kathryn Willan, Ellena Badrick, Gillian Santorelli, Tiffany Yang, Bo Hou, Andrew Steptoe, Giorgio Di Gessa, Jingmin Zhu, Paola Zaninotto, Angela Wood, Genevieve Cezard, Samantha Ip, Tom Bolton, Alexia Sampri, Elena Rafeti, Fatima Almaghrabi, Aziz Sheikh, Syed A Shah, Vittal Katikireddi, Richard Shaw, Olivia Hamilton, Michael Green, Theocharis Kromydas, Daniel Kopasker, Felix Greaves, Robert Willans, Fiona Glen, Steve Sharp, Alun Hughes, Andrew Wong, Lee Hamill Howes, Alicja Rapala, Lidia Nigrelli, Fintan McArdle, Chelsea Beckford, Betty Raman, Richard Dobson, Amos Folarin, Callum Stewart, Yatharth Ranjan, Jd Carpentieri, Laura Sheard, Chao Fang, Sarah Baz, Andy Gibson, John Kellas, Stefan Neubauer, Stefan Piechnik, Elena Lukaschuk, Laura C Saunders, James M Wild,

Stephen Smith, Peter Jezzard, Elizabeth Tunnicliffe, Zeena-Britt Sanders, Lucy Finnigan, Vanessa Ferreira, Mark Green, Rebecca Rhead, Milla Kibble, Yinghui Wei, Agnieszka Lemanska, Francisco Perez-Reche, Dominik Piehlmaier, Lucy Teece, Edward Parker.

**Contributors**  Conceptualisation: AW, LAT, JM, FG and SD; Data curation: ALS, RYP, AW, SCJB, ID, CB and BG; Formal analysis: ALS, RYP, AW, JT; Funding acquisition: BG; Investigation: ALS, AW and RYP; Methodology: ALS, AW, LAT, RYP, JM, FG, JT and KB; Project administration: AW, LAT, AM and BG; Resources: AW, ID, SCJB, BG; Software: ALS, RYP, AW, JT, SCJB, ID and CB; Supervision: AW, LAT, BM, AM and BG; Validation: ALS, RYP and AW; Visualisation: ALS, RYP, AW; Writing-original draft: ALS, RYP and AW; Writing-review and editing: all authors. AW is the guarantor. All authors gave final approval of the version to be published and agreed to be accountable for all aspects of the work.

**Funding**  The OpenSAFELY Platform is supported by grants from the Wellcome Trust (222097/Z/20/Z) and MRC (MR/V015737/1, MC_PC_20059, MR/W016729/1). In addition, development of OpenSAFELY has been funded by the Longitudinal Health and Wellbeing strand of the National Core Studies programme (MC_PC_20030, MC_PC_20059), the NIHR funded CONVALESCENCE programme (COV-LT-0009), NIHR (NIHR135559, COV-LT2-0073), and the Data and Connectivity National Core Study funded by UK Research and Innovation (MC_PC_20058), and Health Data Research UK (HDRUK2021.000, 2021.0157). BG's work on better use of data in healthcare more broadly is currently funded in part by the Wellcome Trust, NIHR Oxford Biomedical Research Centre, NIHR Applied Research Collaboration Oxford and Thames Valley, the Mohn-Westlake Foundation; all University of Oxford, Bennett Institute for Applied Data Science staff are supported by BG's grants on this work. RYP is supported by the EPSRC Centre for Doctoral Training in Health Data Science (EP/S02428X/1). FG is supported by the NIHR Applied Research Collaborative for North West London. SD is supported by: (a) the BHF Data Science Centre led by HDR UK (grant SP/19/3/34678), (b) NIHR University College London Hospitals Biomedical Research Centre, (c) BHF Accelerator Award (AA/18/6/24223), d) the NIHR-UKRI CONVALESCENCE study. KB is funded by a Wellcome Senior Research Fellowship (220283/Z/20/Z).

**Disclaimer**  The views expressed are those of the authors and not necessarily those of the NIHR, NHS England, Public Health England or the Department of Health and Social Care. Funders had no role in the study design, collection, analysis, and interpretation of data; in the writing of the report; and in the decision to submit the article for publication.

**Competing interests**  Over the past 5 years, BG has received research funding from the Laura and John Arnold Foundation, the NHS National Institute for Health Research (NIHR), the NIHR School of Primary Care Research, the NIHR Oxford Biomedical Research Centre, the Mohn-Westlake Foundation, NIHR Applied Research Collaboration Oxford and Thames Valley, the Wellcome Trust, the Good Thinking Foundation, Health Data Research UK (HDRUK), the Health Foundation and the WHO; he also receives personal income from speaking and writing for lay audiences on the misuse of science. CB is an employee of TPP. BM is also employed by NHS England working on medicines policy and clinical lead for primary care medicines data.

**Patient and public involvement**  Patients and/or the public were involved in the design, or conduct, or reporting, or dissemination plans of this research. Refer to the Methods section for further details.

**Patient consent for publication**  Not applicable.

**Ethics approval**  This study involves human participants and was approved by the Health Research Authority (Research Ethics Committee reference 20/L0/0651) and by the London School of Hygiene and Tropical Medicine Ethics Board (reference 21863). Consent was waived as the study was deemed to be low risk.

**Provenance and peer review**  Not commissioned; externally peer reviewed.

**Data availability statement**  No data are available. We used primary care data from approximately 40% of the English population currently registered with GP surgeries using TPP SystmOne software. All data were linked, stored and analysed securely using the OpenSAFELY platform, https://www.opensafely.org/, as part of the NHS England OpenSAFELY COVID-19 service. Data include pseudonymised data such as coded diagnoses, medications and physiological parameters. No free text data are included. All code is shared openly for review and re-use under MIT open license (https://github.com/opensafely/long-covid-sick-notes). Similarly, pseudonymised datasets including ONS registered deaths, hospital episode statistics (HES) and Second Generation Surveillance System (SGSS) COVID-19 test results are securely provided to TPP and linked to primary care data. Detailed pseudonymised patient data are potentially reidentifiable and not shared.

**ORCID iDs**
Andrea L Schaffer http://orcid.org/0000-0002-3701-4997
Ben Goldacre http://orcid.org/0000-0002-5127-4728
Laurie A Tomlinson http://orcid.org/0000-0001-8848-9493
Alex Walker http://orcid.org/0000-0003-4932-6135

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
