## [Reviewer comments · BMJ Open]

ARTICLE DETAILS

TITLE (PROVISIONAL)	Changes in sick notes associated with COVID-19 from 2020 to 2022: A cohort study in 24 million primary care patients in OpenSAFELY-TPP
AUTHORS	Schaffer, Andrea; Park, Robin Y; Tazare, John; Bhaskaran, Krishnan; MacKenna, Brian; Denaxas, S; Dillingham, Iain; Bacon, Seb; Mehrkar, Amir; Bates, Christopher; Goldacre, Ben; Greaves, Felix; Macleod, John; Consortium, The OpenSAFELY; Collaborative, National Core Studies; Tomlinson, Laurie; Walker, Alex

VERSION 1 – REVIEW

REVIEWER	Yilmaz, Canan Tuz Bursa Uludag Universitesi, Family Medicine
REVIEW RETURNED	04-Nov-2023

GENERAL COMMENTS	Congratulations on your fantastic work on this manuscript. a Your manuscript is fantastic as well as clear and smooth while reading. I have two concerns: 1. the third outcome: Is it possible to explain the reason for comparing people hospitalized with pneumonia with a sentence? (For instance why you didn't compare people in the intensive care units for respiratory problems?)2. When we talk about fit notes, it is a term for working status; and you have already mentioned in reference 15; that the data was found with ages between; 18-24/ 25-49/ 50-64. I would recommend renewing the Table 2 ages part with this information but if you have a brief reason to give the data by 18-24/25-34/35-44/45-54/55-64 I would like to hear.
--

REVIEWER	Blanco, Sara Ares Servicio Madrileño de Salud
REVIEW RETURNED	21-Jan-2024

GENERAL COMMENTS	Thanks for your interesting work. I think the article is interesting and well-written. From my perspective, I recommend its publication with minor changes. I would appreciate it if you could review some aspects: Abstract: - I don't understand the meaning the authors intend to convey with the phrase "long-term sickness absence from employment has negative economic consequences." Considering that BMJ Open is an internationally read journal, it's essential to explain the legal and economic framework of long-term sickness in the article. Not all European countries impose economic penalties on patients on fit note. If this phrase is interpreted within the economy of a country, an alternative analysis and the use of other variables not covered in this study might be warranted.
--

- It would be crucial to cite some Hazard Ratios (HRs) for 0-150 days, as this is a key aspect of the study and should be highlighted.
- The conclusions do not align with the results.

Background:

- As mentioned in the abstract, a thorough explanation of how the fit note and the sick leave system works in the UK is recommended. In many European countries there is no difference between fit note and sick note related to COVID-19. It is essential to describe the difference in UK. This includes detailing the possible type of benefits during fit notes, the duration of these benefits, and what happens when they are exhausted.
- It would be insightful to understand the impact of fit notes due to COVID-19 compared to fit notes for other conditions to gauge the magnitude of COVID-19 on healthcare and its ability to exacerbate other health inequalities.
- Regarding the characteristics of fit notes due to COVID-19, citing international studies describing fit notes or sick leaves due to long COVID-19 would be beneficial.

Methods:

- I'm uncertain about the choice of pneumonia as a pathology for comparison with COVID-19, as most pneumonias are bacterial, and the post-viral symptom presentation of COVID-19 and other viruses (EBV, CMV, etc.) is dissimilar. Justification and a comparison of complication percentages, severity, and the need for oxygen in bacterial pneumonias should be provided to assess the variable's validity. Additionally, it's essential to clarify whether bacterial pneumonia also requires fit note from the first day of absence or is processed from the 7th day, as this is crucial for a proper understanding of the results.
- Comparing 11 cohorts might be confusing; a graphical representation could enhance clarity and simplify the text.
- I don't understand why the cutoff point ends at 150 days and not longer. Explaining the rationale behind this decision is crucial since one of the strengths of this work is quantifying the impact of long COVID-19 over time.

Results

- Table 1: The table is too extensive; it should be moved to the supplement, and a simplified version should be presented in the original article, including age, sex, IMD, and reducing other values, especially health conditions with less than 3% prevalence.
- Table 2: It might be more interesting to include comorbidities rather than regions. The section on regions could be reserved for another article aimed at a more local audience.
- Table 3: While I appreciate the authors' transparency in describing crude HR, age-sex adjusted HR, and fully adjusted HR*, this table could be placed in the supplement, and in the original article, only the fully adjusted HR* should be retained to maintain consistency with Figure 1.
- Figure 1: The graph lacks an "s" in "COVID-19 cohort" in the first figure. If the region is removed, it should also be excluded from this figure.
- Supplement Table 2: This table should be included in the original article, replacing regions with comorbidities. It would be helpful if the follow-up time for 0-30, 0-90, and 0-150 days could be included in

	this table. Discussion:  - In the summary, please focus on summarizing results. The last sentence represents an opinion rather than a result. - The explanation of findings in context makes more sense in the introduction to help readers understand the work. It's relevant to explain why there are fewer COVID-19 diagnoses in 2020 compared to 2021 and 2022. - There's a noticeable increase in the total population between 2019 and 2022; an explanation for this should be provided. - The discussion should include a thorough analysis and comparison of the results with other studies. While the results section is extensive and is better presented in tables, understanding the implications of the findings requires more discussion. Specifically, the duration of fit note from 0-150 days needs more attention. Are the UK data similar to other European countries? The results section should be condensed, and the findings should be discussed more thoroughly. - Upon reviewing the cohorts, it's striking that the pneumonia cohort is older, has more smokers, and more frequently has other respiratory diseases. This should be addressed in the discussion, as the longer fit notes duration might be due to the baseline comorbidities of patients who may be in a worse clinical condition, rather than the pneumonia itself. - The most critical results, in my view, are: 1. Many patients require fit notes in the first month. 2. Fit notes continue to be necessary between 0-90 and 0-150 days. 3. Fit notes affect more to women and socioeconomically deprived individuals. 4. COVID-19 increases the likelihood of needing a fit note compared to other illnesses in the general population. - The policy implications should focus on reducing inequalities in women (it would be interesting to see fit notes by age in women to identify the most vulnerable group to COVID-19) and in socioeconomically deprived population groups. In my opinion, this is more critical than references to the economy in the text. - Practical implications that need discussion: Who bears the workload of issuing fit notes (primary care?)? How many consultations are dedicated to fit notes? These questions lead to the next one: What do these results mean for the NHS, and what other activities are compromised due to the high number of COVID-19 patients needing fit notes? - And finally, your work plays a crucial role in quantifying the impact of COVID-19 on people's lives. It is imperative to reassess public health measures, particularly those concerning the assurance of air quality (using HEPA filters, improving natural ventilation, etc.), in order to prevent transmission and minimize the number of infections.
--	--

REVIEWER	Astier-Peña, Maria-Pilar Catalan Institute of Health, Territorial Healthcare Quality Unit. Primary Care
REVIEW RETURNED	03-Mar-2024

GENERAL COMMENTS	The study describes the fit notes related to COVID-19 infection during the 3 years of the pandemic (2020,2021,2022). The trend on fit notes decreased over the years as the vaccination roll outreach
---

more people. Considering hospitalized patients, the fit notes were longer. Nevertheless, it is difficult to analyze the burden of long-term COVID-19 on the infected population as fit notes may finish if the worker is fired.

The research appears comprehensive and well-structured, addressing the impact of SARS-CoV-2 infection and recovery on the workforce by analyzing fit note data in primary care. However, there are some suggestions for improvement:

Title:

I would suggest shortening the title. As a suggestion:

The abstract is adequate.

Keyword: there is an incongruence among keywords in the general information at the beginning and in the text:

Keyword front page:

COVID-19, EPIDEMIOLOGY, PUBLIC HEALTH, Primary Care < Primary

Health Care

Keyword in the manuscript:

sars-cov-2, covid-19, long covid, fit notes, sick notes, sick leave

I would suggest making a mix as follows: COVID-19, sick leave, primary care, epidemiology, public health

The Clarity in Objectives:

The objectives are stated clearly but could be more specific. For instance, defining what is considered a fit note in the UK (people entitled to have it, if it has a weekly or monthly payment, or if it is only to avoid loose a job). How many days a person can have a fit note in England? All these issues may affect the discussion and conclusions.

Long COVID Definition:

Given the heterogeneous nature of long-term COVID-19, the paper acknowledges its difficulty in definition. It might be beneficial to include a brief discussion or reference on the evolving understanding of COVID and its impact on workability. Could you propose a diagnosis code to incorporate into ICPC 2 or ICD 10 to involve doctors in being more accurate on diagnosis?

It would also be of interest to know the rate of people with a fit note in the 3 years of study with the same diagnosis of COVID-19.

Comparison Groups:

I do agree with the research comparing fit note rates among COVID-19 cohorts and general population cohorts. They provide information on the matching process and potential confounders which gives strength and robustness to the study.

Fit Note Rate Interpretation in the discussion section:

The interpretation of fit note rates could be more explicit. For instance, discussing the implications of the decreasing fit note rates over time and their correlation with the evolving understanding of COVID-19 could add depth to the analysis.

Limitations Section:

While the study mentions limitations related to the coding of long COVID, discussing potential biases, such as the impact of the type of fit note selected or the type of contract the patient has with her/his company or the limitation of the period for a fit note and their impact on the results would contribute to a more comprehensive understanding of the study's limitations.

Reproducibility:

I think that with the information provided as supplement material reproducibility is feasible with the data management process and statistical analyses. Authors include additional details on variable definitions, data cleaning steps, and statistical procedures to facilitate transparency and reproducibility. Very useful supplement material for readers.

Patient and Public Involvement:

The involvement of patients and the public is a positive aspect. The

	authors have a publicly available website https://opensafely.org/ through which they invite any patient or member of the public to contact regarding this study: OpenSAFELY project. However, specifying if the public has sent any comment or any trade union has contacted researchers can add value to the information. Discussion Section The discussion could be expanded to include potential policy implications, recommendations for further research, and the broader societal impact of the findings would be of interest. Particularly, in the introduction section authors add information regarding who is entitled to have a fit note and which are the work guarantees of a sick note. At the same time as general practitioners, we have to clearly state that fit notes are a therapeutic tool. The indication for a fit note is a medical issue and the follow-up is essential to guarantee a recovery. In case of not having the expected effect on recovery, think about the possibility of informing on the need for a job change due to the impact of the disease. Visual Aids: The figures and tables to represent key findings or trends are very well performed as there are lot of information to analyze. The availability of supplement materials is very useful and enhances the reader's understanding and engagement. These suggestions aim to refine specific aspects of the research and enhance its overall clarity and impact.
--	---

VERSION 1 – AUTHOR RESPONSE

Reviewer:1

Dr.CananTuzYilmaz,BursaUludagUniversitesi

Comments to the Author:

Congratulations on your fantastic work on this manuscript.

Your manuscript is fantastic as well as clear and smooth while reading.

Thank you for the comments.

I have two concerns:

1. the third outcome: Is it possible to explain the reason for comparing people hospitalized with pneumonia with a sentence? (For instance why you didn't compare people in the intensive care units for respiratory problems?)

We chose pneumonia because like COVID-19, it is commonly the result of a respiratory infection that can result in hospitalisation. It also provides context, as it is a common, serious respiratory disease that clinicians and policy makers will be familiar with. We did not include respiratory problems as a whole, as this would be quite heterogeneous, and include non-infectious conditions such as COPD or asthma, and pneumonia is one of the most common reasons for hospitalisation due to respiratory conditions

<https://digital.nhs.uk/data-and-information/publications/statistical/hospital-admitted-patient-care-activity/2021-22>). We have now clarified this in the text:

Methods, Page 5: *“We chose pneumonia as a comparator to provide context, as it is a common, serious respiratory disease with which clinicians and policy makers will be familiar.”*

2. When we talk about fit notes, it is a term for working status; and you have already mentioned in reference 15; that the data was found with ages between; 18-24/ 25-49/ 50-64. I would recommend renewing the Table 2 ages part with this information but if you have a brief reason to give the data by 18-24/25-34/35-44/45-54/55-64 I would like to hear.

We chose these age categories a priori, so that we had sufficient granularity to understand how the effects differed by age. While ONS report employment data based on the age categories suggested by the reviewer, we don't believe that there would be a benefit in using those age categories in our analysis as well.

Reviewer:2

Mrs. Sara Ares-Blanco, Servicio Madrileño de Salud, Investigation Support Multidisciplinary Unit for Primary Health care and Community North Area of Madrid, World Organization of Family Doctors, Servicio Madrileño de Salud, University of Ljubljana, Tel Aviv University, World Organization of Family Doctors, World Organization of Family Doctors, Luxembourg University, WONCA World (Global Family Doctors) Board Member

Comments to the Author:

Thanks for your interesting work. I think the article is interesting and well-written. From my perspective, I recommend its publication with minor changes. I would appreciate it if you could review some aspects:

Thank you for your comments.

Abstract:

-I don't understand the meaning the authors intend to convey with the phrase "long-term sickness absence from employment has negative economic consequences." Considering that BMJ Open is an internationally read journal, it's essential to explain the legal and economic framework of long-term sickness in the article. Not all European countries impose economic penalties on patients on fit note. If this phrase is interpreted within the economy of a country, an alternative analysis and the use of other variables not covered in this study might be warranted.

Thank you for the suggestion. Given the vagueness of that statement, we have now removed it from the Abstract. To help international readers better understand the context, we have included a few more details about fit notes and citation for those who want more information. We now also use the term "sick notes" instead of fit notes as we believe this will be more understandable.

Methods, Page 6: *"In England, employees can self-certify for the first seven days of sickness; after this, they can receive a Statement of Fitness for Work (also called a fit note or sick note) from their general practitioner (GP) if they determine that the patient's health affects their fitness for work. Unemployed patients can also receive a sick note to support claims for health-related benefits. Sick notes can initially be issued for up to 3 months, and then periodically reviewed if needed. During this time, people can receive statutory sick pay. (11)*

We identified the first recorded sick note over patient follow-up. Sick notes were identified using Clinical Terms Version 3 (CTV3) codes. Sick notes issued due to COVID-19 did not differ from sick notes issued at other times or for other health conditions. People who were required to isolate beyond seven days due to having

or living with someone with symptoms of COVID-19 would be issued an isolation note, rather than a sick note. Isolation notes did not require contact with a GP and are not counted.”

-It would be crucial to cite some Hazard Ratios (HRs) for 0-150 days, as this is a key aspect of the study and should be highlighted.

We believe there is a misunderstanding. Our primary analysis used the entire follow-up period for each year (up to 10 months). We also performed secondary analyses, using different cutoffs (30 days, 90 days and 150 days) to better understand the timing of the increased risk of fit notes. Therefore, the HRs reported in the Abstract are in fact for the entire follow-up period.

We have now made this more explicit both in the Abstract, and in the Methods, and used clearer language in the relevant tables:

Methods, Page 7: *“To determine if the relative differences in sick note occurrence changed over time, we estimated crude and adjusted HRs and 95% CIs both over the entire follow-up period (up to 10 months) and censoring follow-up at 30, 90, and 150 days, and examined whether HRs changed according to the duration of follow-up.”*

-The conclusions do not align with the results.

We have now rewritten the Conclusion as follows:

Abstract: *“Given the under-recording of post-acute COVID-19-related symptoms, these findings contribute a valuable perspective on the long-term effects of COVID-19. Despite likely underestimation of the sick note rate, sick notes were*

issued more frequently to people with COVID-19 compared to those without, even in an era when most people are vaccinated. Most sick notes occurred in the first 30 days post-diagnosis, but the increased risk several months post-diagnosis may provide further evidence of the long-term impact.”

Background:

- As mentioned in the abstract, a thorough explanation of how the fit note and the sick leave system works in the UK is recommended. In many European countries there is no difference between fit note and sick note related to COVID-19. It is essential to describe the difference in UK. This includes detailing the possible type of benefits

during fit notes, the duration of these benefits, and what happens when they are exhausted.

As mentioned above, we have now included further information in the Methods about the nature of fit notes in the UK, along with a citation for people who want more information.

- It would be insightful to understand the impact of fit notes due to COVID-19 compared to fit notes for other conditions to gauge the magnitude of COVID-19 on healthcare and its ability to exacerbate other health inequalities.

We agree that this would potentially be interesting. Unfortunately, the reason that a fit note was issued is not well recorded in the data. Second, long-term symptoms may not necessarily be recognised as being related to COVID-19, especially if the symptoms are vague (e.g. fatigue). Therefore, there would likely be substantial misclassification as to whether a fit note was COVID-19-related or not. However, as one of our secondary analyses we did look at fit notes issued to people hospitalised with pneumonia for comparison.

- Regarding the characteristics of fit notes due to COVID-19, citing international studies describing fit notes or sick leaves due to long COVID-19 would be beneficial.

We have now added a paragraph to the Discussion comparing our work to studies from other jurisdictions:

Discussion, Page 11: *“Direct comparisons with studies of sickness absences in other jurisdictions are difficult due to differences in study populations, follow-up periods, and how sick leave is defined. In a large Swedish study (n=661,780), 6% experienced long-term sick leave associated with COVID-19, more common in women, older people, and people with lower incomes, more comorbidities, or hospitalised with COVID-19. (21) Other studies focussed either on people with COVID-19 only, or certain populations. In Wales, 15% of domiciliary care workers (n=15,931) were issued a sick note between March 2020 and November 2021, and they were more common in women and older people. (22) In Germany, a study of 30,950 people with COVID-19 found that 6% experienced long-term sick leave. (23)”*

Methods:

- I'm uncertain about the choice of pneumonia as a pathology for comparison with COVID-19, as most pneumonias are bacterial, and the post-viral symptom presentation of COVID-19 and other viruses (EBV, CMV, etc.) is dissimilar. Justification and a comparison of complication percentages, severity, and the need for oxygen in bacterial pneumonia should be provided to assess the variable's validity. Additionally, it's essential to clarify whether bacterial pneumonia also requires fit notes from the first day

of absence or is processed from the 7th day, as this is crucial for a proper understanding of the results.

Pneumonia was chosen because like COVID-19, it is commonly the result of a respiratory infection that can result in hospitalisation. Even though the two conditions differ in certain ways, including it for comparison provides context, as it is a common, serious respiratory disease that clinicians and policymakers will be familiar with. Therefore it is not necessary that the two conditions be identical as it is intended to serve as a benchmark, not a control group.

All fit notes are issued from the 7th day, regardless of reason. This is now clarified in the Methods:

Methods, Page 6: *"In England, employees can self-certify for the first seven days of sickness; after this, they can receive a Statement of Fitness for Work (also called a fit note or sick note) from their general practitioner (GP) if they determine that the patient's health affects their fitness for work. Unemployed patients can also receive a sick note to support claims for health-related benefits. Sick notes can initially be issued for up to 3 months, and then periodically reviewed if needed. During this time, people can receive statutory sick pay. (11)... Sick notes issued due to COVID-19 did not differ from sick notes issued at other times or for other health conditions.*

- Comparing 11 cohorts might be confusing; a graphical representation could enhance clarity and simplify the text.

We have now added a figure to the manuscript, depicting the various comparisons in our analysis:

- I don't understand why the cutoff point ends at 150 days and not longer. Explaining the rationale behind this decision is crucial since one of the strengths of this work is quantifying the impact of long COVID-19 over time.

As described above, there was a mis-understanding and our primary analysis was over the entire follow-up period (upto 10 months), not only 150 days. We have now clarified this in the Methods.

Results

- Table 1: The table is too extensive; it should be moved to the supplement, and a simplified version should be presented in the original article, including age, sex, IMD, and reducing other values, especially health conditions with less than 3% prevalence.

We have moved conditions with low prevalence to a Supplementary Table.

However, we would prefer to keep Region in this table, as it is correlated with socioeconomic status in England and hence there was substantial variation in the fit note rate in different regions which we believe is an interesting finding. Also, we have included Region in all of our subsequent result tables/figures. Therefore, it would be unusual not to also include Region in Table 1.

If the reviewer or editors prefer, we can move Region to the Supplementary Files, but for consistency this would involve also removing Region from all subsequent tables/figures, not just Table 1.

- Table 2: It might be more interesting to include comorbidities rather than regions. The section on regions could be reserved for another article aimed at a more local audience.

Thank you for the suggestion. As stated above, we believe that Region is an important factor in predicting fit note receipt.

We would prefer not to present rates by comorbidities as first, most comorbidities had very low rates. Second, the aim of this study was to focus more on the broader societal impacts of COVID-19 and fit notes, rather than in specific clinical groups. As such, this analysis was not specified a priori and we did not include an analysis stratified by comorbidity status in our protocol.

- Table 3: While I appreciate the authors' transparency in describing crude HR, age-sex adjusted HR, and fully adjusted HR*, this table could be placed in the supplement, and in the original article, only the fully adjusted HR* should be retained to maintain consistency with Figure 1.

We have now shifted the non-fully adjusted HRs to the Supplementary Files.

- Figure 1: The graph lacks an "s" in "COVID-19 cohort" in the first figure. If the region is removed, it should also be excluded from this figure.

Thank you for pointing this out, we have now modified the Figure.

- Supplement Table 2: This table should be included in the original article, replacing regions with comorbidities. It would be helpful if the follow-up time for 0-30, 0-90, and 0-150 days could be included in this table.

We prefer to keep Table 2 (which presents the fit note rate per 100 person-months) in the main body of the text, and Supplementary Table 2

(percentage of people receiving a fit note) in the Supplementary Files. The percentages in the Supplementary table do not take into account that different people have a different amount of follow-up time, whereas Table 2 does. We believe Supplementary Table 2 provides useful context, but should not be considered the main result.

Discussion:

- In the summary, please focus on summarizing results. The last sentence represents an opinion rather than a result.

We have now removed that sentence from the summary and shifted it elsewhere in the Discussion.

- The explanation of findings in context makes more sense in the introduction to help readers understand the work. It's relevant to explain why there are fewer COVID-19 diagnoses in 2020 compared to 2021 and 2022.

We have now modified the Introduction as follows (new text in bold):

Introduction, Page 4: *"The pandemic has evolved over time, both in the number of cases, disease severity, and demographic groups most affected. (7) Given the risk of long-term symptoms following COVID-19 ("long COVID"), it is of public health interest to quantify the impact of infection and recovery on the workforce and how that has changed over the pandemic."*

- There's a noticeable increase in the total population between 2019 and 2022; an explanation for this should be provided.

We think this is a misunderstanding. The general population cohorts were matched (upto 10 people) to the COVID-19 cohorts on age, sex and region. As the number of people diagnosed with COVID-19 increased, so did the number of people in the general population cohorts. They do not represent the entire general population. We have added text to the title of Table 1 to make this clearer.

- The discussion should include a thorough analysis and comparison of the results with other studies. While the results section is extensive and is better presented in tables, understanding the implications of the findings requires more discussion. Specifically, the duration of fit note from 0-150 days needs more attention. Are the UK data similar to other European countries? The results section should be condensed, and the findings should be discussed more thoroughly.

We have slightly reduced the text in the Results. But, given the number of results in our study, we believe the amount of text in the Results section is required to provide context and help readers interpret the tables and figures appropriately. However, we have also now added a paragraph comparing our findings to other jurisdictions:

Discussion, Page 11: "Direct comparisons with studies of sickness absences in other jurisdictions are difficult due to differences in study populations, follow-up periods, and how sick leave is defined. In a large Swedish study (n=661,780), 6% experienced long-term sick leave associated with COVID-19, more common in women, older people, and people with lower incomes, more comorbidities, or hospitalised with COVID-19. (21) Other studies focussed either on people with COVID-19 only, or certain populations. In Wales, 15% of domiciliary care workers (n=15,931) were issued a sick note between March 2020 and November 2021, and they were more common in women and older people. (22) In Germany, a study of 30,950 people with COVID-19 found that 6% experienced long-term sick leave. (23)"

- Upon reviewing the cohorts, it's striking that the pneumonia cohort is older, has more smokers, and more frequently has other respiratory diseases. This should be addressed in the discussion, as the longer fit notes duration might be due to the baseline comorbidities of patients whom may be in a worse clinical condition, rather than the pneumonia itself.

While the pneumonia cohort is older and sicker, we have adjusted for all demographic and clinical characteristics in our Cox regression. The fact that the pneumonia cohort is older/sicker is why the crude HR is higher than the fully adjusted HR in Supplementary Table 4. We have added this to the Discussion:

Discussion, Page 11: "Although people hospitalised with pneumonia were older and had more comorbidities, after adjustment for demographics and clinical

characteristics people hospitalised with COVID-19 were still less likely to be issued a sick note than people with pneumonia, especially in 2022 suggesting the

long-term effects are not worse than comparable serious respiratory infections requiring hospitalisation.”

- The most critical results, in my view, are: 1. Many patients require fit notes in the first month. 2. Fit notes continue to be necessary between 0-90 and 0-150 days. 3. Fit notes affect more to women and socioeconomically deprived individuals. 4. COVID-19 increases the likelihood of needing a fit note compared to other illnesses in the general population.

Thank you for the suggestion, we have made modifications as described below.

- The policy implications should focus on reducing inequalities in women (it would be interesting to see fit notes by age in women to identify the most vulnerable group to COVID-19) and in socioeconomically deprived population groups. In my opinion, this is more critical than references to the economy in the text.

We have now added a paragraph discussing the implications of our results on people living in deprived areas, and in women and other high risk groups:

Discussion Page 12: “In the UK, the sickness absence rate in 2022 is the highest it has been since 2004. (32) The populations with the highest absolute rates of sick notes after COVID-19 (women, people living in areas of greater deprivation) are those over-represented in low-income and/or public-facing jobs, which have higher rates of sickness absences. (33) They also have high rates of mental health problems, which are associated both with long-term sickness absence (1, 34) and long COVID (31), likely pointing to unmet needs. However, although people living in areas of least deprivation with COVID-19 had the lowest absolute sick note rate, they had the greatest relative increase compared with the general population in earlier years, reflecting their lower baseline rate; this pattern disappeared in 2022.”

- Practical implications that need discussion: Who bears the workload of issuing fit notes (primary care)? How many consultations are dedicated to fit notes? These questions lead to the next one: What do these results mean for the NHS, and what other activities are compromised due to the high number of COVID-19 patients needing fit notes?

These are interesting questions, which cannot be answered in this work. We have now added the following to the Discussion. We also currently have a grant to do work into pressures on primary care, and this type of research question would fit well within that body of work.

Discussion Page 12: “Pressures on primary care are also increasing, and over the past decade the number of GPs has gone down and each GP is thus responsible

for a greater number of patients.(35) It's unclear to what extent the increasing number of people experiencing long-term post-acute symptoms contributes to the burden on primary care. Thus, we need to understand not only how people with long COVID symptoms interact with primary care, but how their symptoms are recorded in electronic health records to inform future studies."

- And finally, your work plays a crucial role in quantifying the impact of COVID-19 on people's lives. It is imperative to reassess public health measures, particularly those concerning the assurance of air quality (using HEPA filters, improving natural ventilation, etc.), in order to prevent transmission and minimize the number of infections.

Thank you for the suggestions, we will make note of this for future work.

Reviewer:3

Dr. Maria-Pilar Astier-Peña, Catalan Institute of Health, Fundació IIS Aragón

Comments to the Author:

The study describes the fit notes related to COVID-19 infection during the 3 years of the pandemic (2020, 2021, 2022). The trend on fit notes decreased over the years as the vaccination roll outreach more people. Considering hospitalized patients, the fit notes were longer. Nevertheless, it is difficult to analyze the burden of long-term COVID-19 on the infected population as fit notes may finish if the worker is fired.

The research appears comprehensive and well-structured, addressing the impact of SARS-CoV-2 infection and recovery on the workforce by analyzing fit note data in primary care. However, there are some suggestions for improvement:

Thank you for the comments.

Title:

I would suggest shortening the title. As a suggestion:

Based on the editor's suggestion, we have kept the title as is.

The abstract is adequate.

Keyword: there is an incongruence among keywords in the general information at the beginning and in the text:

Keyword front page:

COVID-19, EPIDEMIOLOGY, PUBLIC HEALTH, Primary Care < Primary

HealthCare

Keyword in the manuscript:

sars-cov-2, covid-19, long covid, fit notes, sick notes, sick leave

I would suggest making a mix as follows: COVID-19, sick leave, primary care, epidemiology, public health

This discrepancy is due to different keywords being entered during the submission process. We have now included some of the suggested keywords, and will make sure we have entered them correctly.

The Clarity in Objectives:

The objectives are stated clearly but could be more specific. For instance, defining what is considered a fit note in the UK (people entitled to have it, if it has a weekly or monthly payment, or if it is only to avoid loose a job). How many days a person can have a fit note in England? All these issues may affect the discussion and conclusions.

We have now added more text describing fit notes in England, along with a citation for those who wish to learn more:

Methods Page 6: *"In England, employees can self-certify for the first seven days of sickness; after this, they can receive a Statement of Fitness for Work (also called a fit note or sick note) from their general practitioner (GP) if they determine that the patient's health affects their fitness for work. Unemployed patients can also receive a sick note to support claims for health-related benefits. Sick notes can initially be issued for up to 3 months, and then periodically reviewed if needed. During this time, people can receive statutory sick pay. (11)"*

Long COVID Definition:

Given the heterogeneous nature of long-term COVID-19, the paper acknowledges its difficulty in definition. It might be beneficial to include a brief discussion or reference on the evolving understanding of COVID and its impact on workability. Could you propose a diagnosis code to incorporate into ICPC 2 or ICD 10 to involve doctors in being more accurate on diagnosis?

While a very interesting question, we believe this is out of scope for our work. However, we have included a sentence on this topic under "Future research":

Discussion Page 12: *"Thus, we need to understand not only how people with long COVID symptoms interact with primary care, but how their symptoms are recorded in electronic health records to inform future studies."*

It would also be of interest to know the rate of people with a fit note in the 3 years of study with the same diagnosis of COVID-19.

We are a bit unsure what the reviewer is asking. We'd be happy to respond to this on further clarification.

Comparison Groups:

I do agree with the research comparing fit note rates among COVID-19 cohorts and general population cohorts. They provide information on the matching process and potential confounders which gives strength and robustness to the study.

Fit Note Rate Interpretation in the discussion section:

The interpretation of fit note rates could be more explicit. For instance, discussing the implications of the decreasing fit note rates over time and their correlation with the evolving understanding of COVID-19 could add depth to the analysis.

We have now discussed the reason for the reduced fit note rate in the Discussion:

Discussion Page 10: *"The introduction of vaccination programmes and differences in circulating variants lead to changes in severity and transmissibility of SARS-CoV-2.*

While infection rate went up during the study period, the COVID-19-related mortality rate has decreased in each subsequent wave, in large part due to great vaccination coverage. (7) This reduced severity helps explain the lower sick note rate in later years. "

Limitations Section:

While the study mentions limitations related to the coding of long COVID, discussing potential biases, such as the impact of the type of fit note selected or the type of contract the patient has with her/his company or the limitation of the period for a fit note and their impact on the results would contribute to a more comprehensive understanding of the study's limitations.

The majority of workers (including part-time workers) are eligible to receive fit notes and sick pay. Therefore, we don't believe this is a substantial limitation.

Reproducibility:

I think that with the information provided as supplement material reproducibility is feasible with the data management process and statistical analyses. Authors include additional details on variable definitions, data cleaning steps, and statistical procedures to facilitate transparency and reproducibility. Very useful supplement material for readers.

We believe in open science and reproducible research, and within our manuscript we have a "Software and reproducibility" section where we included a link to our public Github repository, which includes our study protocol, all code used in this study for both data management and analysis, and a list of all the code lists used: <https://github.com/opensafely/long-covid-sick-notes>. A log of every single

analysis run against the data is also available for review. Therefore, as we have included this link we don't believe it is necessary to also include this information in the Supplementary Files.

Patient and Public Involvement:

The involvement of patients and the public is a positive aspect. The authors have a publicly available website <https://opensafely.org> through which they invite any patient or member of the public to contact regarding this study: OpenSAFELY project. However, specifying if the public has sent any comment or any trade union has contacted researchers can add value to the information.

We have not received any specific comments from the public on this work.

Discussion Section

The discussion could be expanded to include potential policy implications, recommendations for further research, and the broader societal impact of the findings would be of interest.

In our Discussion, we have sections entitled both "Policy implications" and "Future research." However, we have also now added a section discussing the meaning of our findings for primary care:

Discussion page 12: *"Pressures on primary care are also increasing, and over the past decade the number of GPs has gone down and each GP is thus responsible for a greater number of patients. (35) It's unclear to what extent the increasing number of people experiencing long-term post-acute symptoms contributes to the burden on primary care. Thus, we need to understand not only how people with long COVID symptoms interact with primary care, but how their symptoms are recorded in electronic health records to inform future studies."*

Particularly, in the introduction section authors add information regarding who is entitled to have a fit note and which are the work guarantees of a sick note. At the same time as general practitioners, we have to clearly state that fit notes are a therapeutic tool. The indication for a fit note is a medical issue and the follow-up is essential to guarantee recovery. In case of not having the expected effect on recovery, think about the possibility of informing on the need for a job change due to the impact of the disease.

We have added further detail about fit notes to the Methods:

Methods page 6: *"In England, employees can self-certify for the first seven days of sickness; after this, they can receive a Statement of Fitness for Work (also called a fit note or sick note) from their general practitioner (GP) if they determine that the*

patient's health affects their fitness for work. Unemployed patients can also receive a sick note to support claims for health-related benefits. Sick notes can initially be issued for up to 3 months, and then periodically reviewed if needed. During this time, people can receive statutory sick pay. (11)"

Also, we agree that people who choose to leave the workforce will not be captured. We have added this as a limitation:

Discussion page 10: *"Last, people with very severe symptoms who need to leave the workforce completely may also not be captured in our data, but these counts are likely to be small."*

Visual Aids:

The figures and tables to represent key findings or trends are very well performed as there are a lot of information to analyze. The availability of supplement materials is very useful and enhances the reader's understanding and engagement.

These suggestions aim to refine specific aspects of the research and enhance its overall clarity and impact.

Thank you for the feedback.

VERSION 2 – REVIEW

REVIEWER	Astier-Peña, Maria-Pilar Catalan Institute of Health, Territorial Healthcare Quality Unit. Primary Care
REVIEW RETURNED	25-May-2024
GENERAL COMMENTS	Thank you to the authors for addressing most of my comments. I respect the decision to retain the regions in Table 1, but I would like to suggest creating a figure with three maps representing the COVID-19 cohorts for 2020, 2021, and 2022 by region. As an international reader, this visual representation would help me understand your perspective and the significance of regions in your analysis. I believe the article is now clearer and should be published in its current version.